# Robot-Assisted Sacrocolpopexy versus Trans-Vaginal Multicompartment Prolapse Repair: Impact on Lower Bowel Tract Function

**DOI:** 10.3390/biomedicines11082105

**Published:** 2023-07-26

**Authors:** Alessia Martoccia, Yazan Al Salhi, Andrea Fuschi, Onofrio Antonio Rera, Paolo Pietro Suraci, Silvio Scalzo, Alice Antonioni, Fabio Maria Valenzi, Manfredi Bruno Sequi, Cosimo De Nunzio, Riccardo Lombardo, Alessandro Sciarra, Giovanni Di Pierro, Giorgio Bozzini, Anastasios D. Asimakopoulos, Enrico Finazzi Agrò, Alessandro Zucchi, Marilena Gubiotti, Mauro Cervigni, Antonio Carbone, Antonio Luigi Pastore

**Affiliations:** 1Urology Unit, Department of Medical-Surgical Sciences and Biotechnologies, Faculty of Pharmacy and Medicine, “Sapienza” University of Rome, 04100 Latina, Italy; martoccia.alessia@gmail.com (A.M.); uroyazan@gmail.com (Y.A.S.); andrea.fuschi@uniroma1.it (A.F.); antoniorera94@gmail.com (O.A.R.); spaolopietro@gmail.com (P.P.S.); silvioscalzo@hotmail.it (S.S.); aliceantonioni@gmail.com (A.A.); fabiovalenzi@gmail.com (F.M.V.); mb.sequi@gmail.com (M.B.S.); mauro.cervigni@libero.it (M.C.); antonio.carbone@uniroma1.it (A.C.); 2Department of Urology, Sant’Andrea Hospital, Sapienza University of Rome, 00189 Rome, Italy; cosimo.denunzio@uniroma1.it (C.D.N.); riccardo.lombardo@uniroma1.it (R.L.); 3Policlinico Umberto I, Department of Urology, Sapienza University of Rome, 00161 Rome, Italy; alessandro.sciarra@uniroma1.it (A.S.); giovannibattista.dipierro@uniroma1.it (G.D.P.); 4Department of Urology, ASST Lariana-Sant’Anna Hospital, 22100 Como, Italy; gioboz@yahoo.it; 5Urology Unit, Fondazione PTV Policlinico Tor Vergata University Hospital, 00133 Rome, Italy; tasospao2003@yahoo.com (A.D.A.); finazzi.agro@med.uniroma2.it (E.F.A.); 6Department of Urology, University of Pisa, 56126 Pisa, Italy; zucchi.urologia@gmail.com; 7Department of Urology, San Donato Hospital, 52100 Arezzo, Italy; marilena.gubiotti@gmail.com

**Keywords:** pelvic organ prolapse, lower bowel tract symptoms, robotic-assisted sacrocolpopexy, transvaginal surgery, patient global impression improvement

## Abstract

Background: This study evaluated the effectiveness, safety, and possible changes in bowel symptoms after multicompartment prolapse surgery by comparing two different surgical approaches, transvaginal mesh surgery with levatorplasty (TVMLP) and robot-assisted sacrocolpopexy (RSC). Methods: All patients underwent pelvic (POP-Q staging system) and rectal examination to evaluate anal sphincter tone in the lithotomy position with the appropriate Valsalva test. The preoperative evaluation included urodynamics and pelvic magnetic resonance defecography. Patient Global Impression of Improvement (PGI-I) at follow-up measured subjective improvement. All patients completed Agachan–Wexner’s questionnaire at 0 and 12 months of follow-up to evaluate bowel symptoms. Results: A total of 73 cases were randomized into the RSC group (36 cases) and TVMLP group (37 cases). After surgery, the main POP-Q stage in both groups was stage I (RCS 80.5% vs. TVMLP 82%). There was a significant difference (*p* < 0.05) in postoperative anal sphincter tone: 35%. The TVMLP group experienced a hypertonic anal sphincter, while none of the RSC group did. Regarding subjective improvement, the median PGI-I was 1 in both groups. At 12 months of follow-up, both groups exhibited a significant improvement in bowel symptoms. Conclusions: RSC and TVMLP successfully corrected multicompartment POP. RSC showed a greater improvement in the total Agachan–Wexner score and lower bowel symptoms.

## 1. Introduction

According to the International Continence Society (ICS), the term pelvic organ prolapse (POP) clinically refers to the descent of one or more sections of the anterior and posterior vaginal wall, the uterine cervix, or the apex of the vagina (vaginal vault or cuff scar after hysterectomy) [1]. POP is identified in up to 50% of women, with heterogeneous symptoms according to the compartment of descent [2]. Women with anterior and apical vaginal wall prolapse demonstrate symptoms related to urinary functions, such as changes in frequency, urgency, and stress urinary incontinence (SUI) or feeling of incomplete evacuation or bladder outlet obstruction [3], while women with posterior vaginal wall prolapse often report symptoms such as obstruction of defecation and incomplete emptying of bowels [4]. Some studies have found a dose-response effect of posterior wall prolapse on lower bowel tract symptoms (LBTS) [5,6]. However, only a few studies [7] have analyzed the effects of surgery on the symptom of obstructed defecation in which resolution or improvement in all bowel symptoms after transvaginal surgery (posterior colporrhaphy, site-specific rectocele repair with or without graft augmentation) have been reported. Most bowel symptoms improved in women with moderate to severe POP after sacrocolpopexy [8].

This study aimed to evaluate the effects, safety, and probable changes in bowel symptoms in patients suffering from prolapse of anterior, apical, and posterior vaginal walls after surgery to correct multicompartment prolapse using two different approaches, transvaginal mesh surgery with levatorplasty (TVMLP) and robot-assisted sacrocolpopexy (RSC) with anterior and posterior mesh.

## 2. Materials and Methods

### 2.1. Study Population and Subdivision

The study was a randomized prospective investigation. The randomization process was created using a computer-generated list of even and odd numbers. The patients were assigned following a simple procedure: even number—TVMLP, odd number—RSC. The trial was non-blinded to both the surgeons and the patients. Patients with multicompartment POP (stage > 2) were included. The study followed the Ethical Principles for Medical Research Involving Human Subjects (World Medical Association, The Declaration of Helsinki Principles, 2000). It was approved by the local ethical committee of Sapienza University Pharmacy and Medicine Faculty, Latina, Italy (no. UnivLSLT.2017/UROICLT20157). All patients signed the informed consent form before enrolling in this study. Two well-experienced surgeons (AC and MC) performed all surgeries. The patients were randomly divided into two groups:Patients selected for RSC with anterior and posterior mesh placement.Patients selected for TVMLP.

### 2.2. Inclusion and Exclusion Criteria

The inclusion criteria included female patients with symptomatic multicompartment prolapse stage III–IV determined according to the Pelvic Organ Prolapse Quantification system: all the patients presented cystocele, hysterocele, and rectocele. The general exclusion criteria were age over 75 years, BMI ≥ 35 kg/m^2^, and any medical condition or psychiatric illness that would render the patient unable to tolerate the post-surgery pain or affect their ability to attend the follow-up visits. According to anterior and middle compartments, all women with previous pelvic surgery were excluded. According to posterior compartment, exclusion criteria included iatrogenic constipation caused by medication, pelvic floor dyssynergia (studied by urodynamics with contextual pelvic electromyography), anismus, and Hirschsprung’s disease, excluded by anal manometry (evaluating maximum anal sphincter pressure in resting and squeezing conditions, as reported in Table 1). Any anorectal abnormality observed by ano-rectal endoscopy examination was excluded.

### 2.3. Patients and Data Collection

All patients were studied preoperatively at time 0 (baseline) and postoperatively at 6 and 12 months. The clinical and demographic characteristics of patients are reported in Table 1. Each patient was evaluated for various criteria, including age, BMI, previous pelvic surgery, and medical history, including prolapse and bowel symptoms.

All patients were preoperatively examined by urogynecologist and colorectal surgeon in the standing and lithotomy positions using leg supports, both during rest and straining. In addition, proctoscopy and a speculum examination were performed. Before surgery, every patient was discussed in a multidisciplinary setting, including colorectal surgeons, urogynecologists, radiologists, and pelvic floor physical therapists.

The preoperative evaluation included pelvic magnetic resonance defecography to assess the presence of rectocele produced by bulging of the anterior wall of the rectum, displacing the posterior wall of the vagina, excluding enterocele, peritoneocele, and rectal intussusception. All patients underwent a pelvic and rectal examination to assess the severity of POP using the POP-Q staging system and to evaluate anal sphincter tone in the lithotomy position with the appropriate Valsalva test, respectively. Considering the absence of a validated classification system for the digital rectal examination in terms of resting pressure, the anal sphincter tone was classified as “hypertonic” when differences between the resting pressure and squeeze pressure were not felt and “normal” when the patient could feel the difference between two pressures. A sacral neurologic examination was performed to evaluate anal and bulbocavernosus reflexes. All the urinary and bowel-related symptoms were recorded in a database. The preoperative workup also included urodynamic examination, urinary tract ultrasound, cervical cytology, and pelvic ultrasound with post-void residual volume. Patient Global Impression of Improvement (PGI-I) was employed to ascertain the subjective improvement at follow-up [9]. All the patients answered Agachan–Wexner’s questionnaire, named Cleveland Clinic Constipation Score [10], at 0 and 12 months to evaluate LBTS, considering a total score between 1–5 “low constipation”, 6–10 “moderate constipation”, 11–15 “high constipation”, and 16–30 “very high constipation”. The choice of Agachan–Wexner Constipation Scoring system was related to its specific evaluation of constipation and obstructed defecation. This scoring system focuses on the presence of unsuccessful attempts and/or straining to defecate, incomplete defecation or multiple times defecation, need for anal digitation, and all other symptoms related to obstructed defecation syndrome. Obstructed defecation syndrome (ODS) was investigated by a general surgeon specialized in coloproctology, who was able to identify the presence of rectocele and external/internal prolapse with a proctological examination. Furthermore, all patients underwent MRI defecography to assess defecatory dynamics and evaluate rectal prolapse and colonoscopy, to exclude cancer or polyps. No statistically significant differences were observed at MRI between the two groups. All patients reported a multicompartmental prolapse with rectocele, excluding any other evidence of anorectal abnormalities. The clinical follow-up involved pelvic and rectal examinations and data collection at 6 and 12 months.

### 2.4. Preoperative Management

All patients had an anesthesia evaluation to determine the perioperative risks and received deep venous thrombosis prophylaxis and perioperative antibiotics [11]. There was no necessity for formal bowel preparation [12].

### 2.5. Surgical Technique

TVMLP was performed in the dorsal lithotomy position. A Foley catheter was placed. A longitudinal incision was performed in the anterior vaginal wall to create the perivesical and pararectal spaces until the pubocervical fascia. The mesh (Figure 1) was fixed using a retractable insertion guide to the obturator membrane and ischiorectal fossa. The anterior vaginal layer was closed [13]. A transverse incision was performed in the posterior vaginal wall to expose the rectovaginal septum. The puborectalis and pubococcygeal muscles were exposed until the posterior fornix. Five or six mattress stitches using nonabsorbable 0 sutures were fixed in the limbs of the muscles to reinforce the rectovaginal septum from the proximal to the distal point. The incision was closed with absorbable sutures.

RSC was performed using the Da Vinci XI surgical system in a three-arm configuration. Hysterectomy was performed only in cases that needed a sub-total hysterectomy to repair the fibroid uterus. After the mobilization of the sigma, the sacral promontory was exposed, and the peritoneum was incised from the sacral promontory to the posterior vaginal fornix, previously exposed with a spatula in the posterior fornix of the vagina. The Levator Ani fascia was exposed, and the posterior mesh (Figure 2) was fixed with two or three nonabsorbable 2/0 stitches. Then, the vaginal spatula was positioned in the anterior fornix, and the plane between the bladder and vagina was developed. The anterior mesh (Figure 2) was fixed with two nonabsorbable 2/0 stitches on the anterior vaginal wall. A running suture with V-lock was passed on the anterior vaginal wall from the base to the apex. If a hysterectomy was not performed, the anterior mesh was passed through to the tunnel, previously conformed in the peritoneum. However, if a hysterectomy was performed, both the meshes were fixed on the sacral promontory with a nonabsorbable 1/0 stitch. The peritoneum was closed with a V-lock running suture.

### 2.6. Postoperative Management

A vaginal gauze was used in all patients for two days after the surgery, and the period of hospital stay depended on each patient’s individual condition. The Foley catheter was removed the day after the surgery. All possible complications were screened for during the follow-up. Intraoperative and perioperative outcomes are summarized in Table 2.

### 2.7. Statistical Analysis

A descriptive statistical analysis was performed for every baseline and postoperative variable. The characteristics of the patients and perioperative outcomes were compared through t-tests. The differences between Agachan–Wexner Score and its sub-scores at baseline and follow-up were evaluated for statistical significance through paired t-tests. Statistical analysis was performed using the Statistical Package for the Social Sciences (SPSS), version 25.0 (SPSS Inc., Chicago, IL, USA). *p*-values < 0.05 were considered statistically significant.

## 3. Results

### 3.1. Comparison of Preoperative and Postoperative Demographic Data

From March 2018 to November 2021, 82 POP-corrective surgeries were performed in our hospital. Of these, 73 cases conforming to the inclusion criteria used in this study were selected. They were then classified into the RSC group (36 cases) and the TVMLP group (37 cases). No significant differences in the demographic data collected at baseline were observed between the two groups. The median age of patients was 66.6 and 66.8 years (*p* = 0.79), and BMI was 21.5 and 22.07 kg/m^2^ (*p* = 0.23) in the RCS and TVMLP groups, respectively. According to the analysis of the preoperative POP-Q stage, stage III was the main stage observed in both the groups (52.7% in the RSC and 51.3% in the TVMLP group), and the patients in both the groups presented cystocele, uterine prolapse, and rectocele (100% in both the RSC and TVMLP groups). The preoperative anal sphincter tone was classified as “normal” in all patients (100% in both the RSC and TVMLP groups), as confirmed by the anal sphincter manometry in rest and squeezing conditions (*p* > 0.05). Post-surgery, the main POP-Q stage in both groups was detected as stage I (80.5% in the RCS and 82% in the TVMLP groups). There was a significant difference (*p* < 0.05) in both the groups concerning postoperative anal sphincter tone: 35% of the patients in the TVMLP group experienced hypertonic anal sphincter. In comparison, none of the patients in the RSC group experienced hypertonic anal sphincter (0%). Regarding subjective improvement in POP, the median PGI-I was 1 in both groups, without any statistically significant differences between the two groups. All the characteristics are summarized in Table 1.

### 3.2. Comparison of Perioperative Parameters

The time consumed for surgery was significantly shorter in the TVMLP group than in the RSC group (mean: 76.62 and 109.35 min; SD: 0.92 and 8.73, respectively; *p* < 0.005) (Table 2), while the bleeding amount was significantly higher in the TVMLP group than in the RSC group (mean: 20.89 and 4.94 mL; SD: 4.44 and1.06; *p* < 0.005). There were no significant differences between the period of hospital stay, rate of post-surgery complications, recurrence of POP, and mesh exposure between the two groups (*p* > 0.005). In both groups, the length of hospital stay was three days as a standard protocol (*p* = 0.31). However, two cases of postoperative incontinence in each group (*p* = 0.09), and no case of mesh exposure, were reported in both groups at 12 months of follow-up.

No significant differences in bowel symptoms between the two groups at the baseline were observed. At follow-up 12 months after surgery, in comparison to the baseline, both groups exhibited a significant improvement in all domains and total score, except for the domain “pain” in the TVMLP group: in which the pain after surgery worsened (pre-surgery: 1.49 and post-surgery: 2.00). The main postoperative differences between the two groups were in favor of RSC group against TVMLP group, especially regarding the domain of pain (mean: 0.50 in RSC group and mean: 2.00 in TVMLP group; SD: 0.50 and 0.97, respectively; *p* < 0.05) and the total Agachan–Wexner Score (mean: 6.88 in RSC group and mean: 8.56 in TVMLP group; SD 1.63 and 1.76, respectively; *p* < 0.05). All data are summarized in Table 3.

## 4. Discussion

In the literature review, it is unclear how POP compromises bladder and bowel functions since these symptoms are not specific only to women with POP [14]. It is commonly known that prolapse is associated with conditions that increase intra-abdominal pressure, such as obesity and pulmonary disease, in addition to aging, obstetric and congenital factors; hence, chronic straining was associated with muscular damage [15,16] to the connective fibers of the pelvic floor, resulting in elongation of the Levator Ani muscle and his hiatus in the sagittal and transverse planes [17,18,19]. It is plausible, considering this abnormal anatomic angle, that prolapse of the posterior vaginal wall interferes with rectal evacuation resulting in the symptom of obstructed defecation. This study aimed to compare RSC and TVMLP surgeries for the reconstruction of pelvic floor reconstruction in women affected by multicompartment prolapse, with a medium-term follow-up to evaluate their effects and safety, with special consideration on LBTS.

Only a few previous studies have compared these two surgical approaches [20,21]. Both these procedures use synthetic meshes to improve curing rates of prolapse and correction of pelvic anatomy from the anterior to the posterior compartment. In this study, both procedures demonstrated an improvement in PGI-I without any significant differences between the two types of surgery.

In this study, intraoperative bleeding was observed to be lesser in the RCS group; robotic surgery is highly advantageous compared to trans-vaginal mesh [22]. Firstly, the use of EndoWrist facilitated the suture passage even to the most difficult-to-reach tissues, such as the Levator Ani fascia. In addition, a 3D-magnified field of view allowed the visualization of even small arteries, due to which the bleeding was lesser; however, in the trans-vaginal approach, the surgery is “blinded”. Hence, robotic surgery can be less invasive and safer than transvaginal.

In this study, the time required for surgery was longer in the RCS group than the TVMLP group, which may reflect upon the recent introduction of robotic surgery in our institution and the longer time required for docking than the surgery time. However, no complications were attributed to the length of surgery, as observed previously [22]. No differences in length of hospital stay and rate of occurrence of complications were observed. Our study indicated that postoperative POP-Q prominently improved anatomic restoration after either RCS or TVMLP without any statistically significant differences between the two types of surgeries. No remarkable differences concerning prolapse recurrence (considering a recurrence with a POP-Q > III after surgery) were observed in our study, as was reported similarly in one such study in which the postoperative recurrence rates of transvaginal and mini-invasive surgery were 8.6% and 5.0% (*p* = 0.064), respectively, with no significant differences [20]. Based on the LBTS data obtained in our study, an improvement in the total Agachan–Wexner score in the RCS group was observed postoperatively than preoperatively. In the TVMLP group, there was also an improvement in the postoperative total score and most postoperative domains. On analysis of each parameter, the pain during defecation after TVMLP was greater than reported during postoperative follow-up. With the post-robotic sacrocolpopexy, many patients experienced improvement or complete resolution in their LBTS, such as constipation with a resolution of pain, as observed in this study. Thus, it can be suggested that sacrocolpopexy prevents redundant sigmoid colon and enterocele pushing down into the pelvis; the meshes and relative peritoneum fibrosis obliterated the deep Pouch of Douglas and eliminated the potential possibility of enterocele, rectocele, and sigmoidocele. In addition, this surgery can straighten the angle of the rectosigmoid junction, facilitating complete defecation. The pain during defecation post-TVMLP may be due to stitches altering the physiological distensibility of the rectum during stool passage and determining a painful hypertonic status of the external anal sphincter, as confirmed through a follow-up digital rectal examination. One of the main limitations of this study was the limited to medium period of follow-up (1 year). Still, this investigation is one of the largest series evaluating the postoperative outcomes and LBTS while comparing two surgical approaches for correcting multicompartment POP.

## 5. Conclusions

RSC and TVMLP successfully corrected multicompartment POP, both observed to be safe procedures, with no differences in duration of hospital stay and rate of occurrence of post-surgery complications. RSC seemed to be less invasive in terms of containing blood loss. In addition, an improvement in the total Agachan–Wexner score was observed in RSC, while TVMLP appeared to be associated with increased pain during defecation.

## Figures and Tables

**Figure 1 biomedicines-11-02105-f001:**
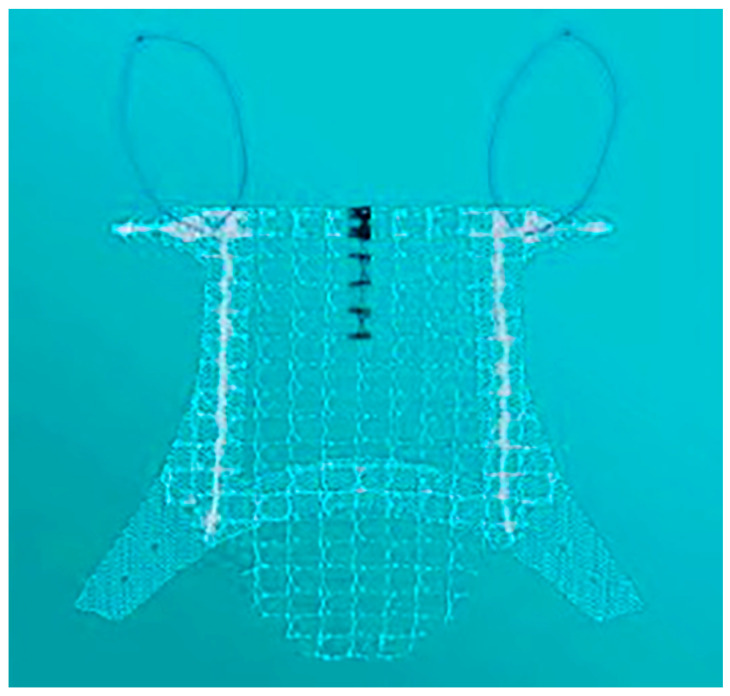
Transvaginal MESH.

**Figure 2 biomedicines-11-02105-f002:**
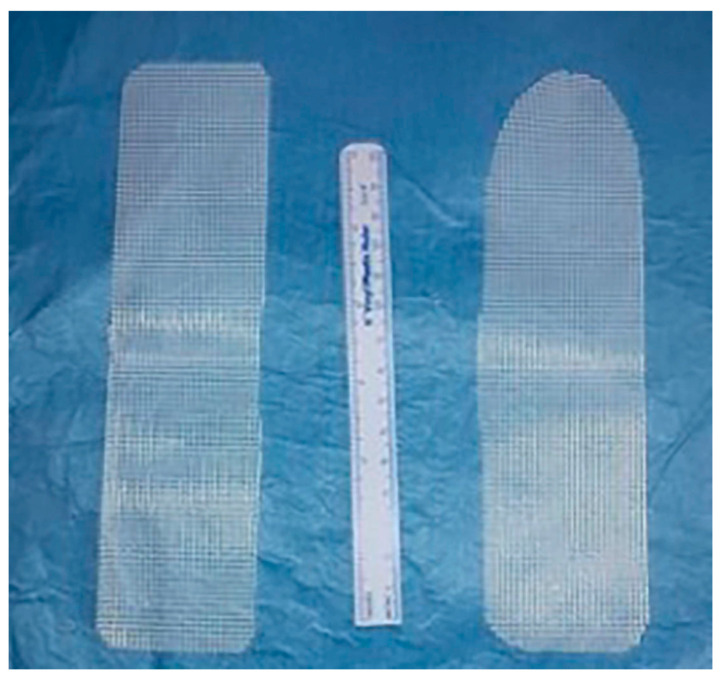
Posterior and anterior meshes for robot-assisted sacrocolpopexy.

**Table 1 biomedicines-11-02105-t001:** Comparison of demographic data (no, mean, SD). SD, standard deviation; PGI-I, patient global impression of improvement; *MARP,* maximum anal resting pressure; MASP, maximum anal squeeze pressure.

	RSC (*n* = 36)	TVMLP (*n* = 37)	*p*-Value
Age, y.o. (SD)	66.6 (3.20)	66.86 (3.17)	0.79
BMI, kg/m^2^ (SD)	21.5 (1.84)	22.07 (2.23)	0.23
Cystocele + uterine prolapse + rectocele (no.)	36	37	
Preoperative POP-Q (*n*, %)			
Stage I	0	0	
Stage II	0	0	
Stage III	19 (52.7%)	19 (51.3%)	0.73
Stage IV	17 (47.2%)	18 (48.6%)	0.56
Postoperative POP-Q (*n*, %)			
Stage I	29 (80.5%)	30 (82%)	0.83
Stage II	7 (17.4%)	7 (18.9%)	0.92
Stage III	0	0	
Stage IV	0	0	
Preoperative hypertonic anal sphincter (no.,%)	0	0	
Postoperative hypertonic anal sphincter (no.,%)	0	13 (35%)	<0.05
Anal sphincter pressure (mmHg)			
MARP (SD)	56.94 ± 11.91	58.84 ± 11.41	>0.05
MASP (SD)	109.86 ± 22.86	111.73 ± 22.81	>0.05
PGI-I scale (SD)	1.10 (0.46)	1.10 (0.52)	1

**Table 2 biomedicines-11-02105-t002:** Comparison of intraoperative and postoperative data (no, mean, SD).

	RSC (*n* = 36)	TVMLP (*n* = 37)	*p*-Value
Operation time, min (SD)	109.35 (8.73)	76.62 (0.92)	<0.05
Bleeding amount, mL (SD)	4.94 (1.06)	20.89 (4.44)	<0.05
Hospital stay, day (SD)	2.9 (0.33)	3 (0.0)	0.31
Complication rate (SD)	0.05 (0.23)	0.05 (0.22)	0.09
Recurrence rate (SD)	0.05 (0.23)	0.05 (0.22)	0.09
Mesh exposure	0	0	

SD, standard deviation.

**Table 3 biomedicines-11-02105-t003:** Comparison of preoperative and postoperative Agachan–Wexner Score (*n*, mean, SD).

	RSC (*n* = 36)	TVMLP (*n* = 37)	*p*-Value
Frequency pre (SD)	1.30 (0.57)	1.29 (0.57)	0.95
Frequency post (SD)	0.55 (0.50)	0.56 (0.50)	0.91
*p*-value	<0.05	<0.05	
Completeness pre (SD)	2.19 (0.88)	2.16 (0.89)	0.87
Completeness post (SD)	1.05 (0.58)	1.29 (0.70)	0.11
*p*-value	<0.05	<0.05	
Difficulty pre (SD)	2.16 (0.91)	2.00 (0.74)	0.39
Difficulty post (SD)	1.05 (0.58)	1.21 (0.67)	0.2
*p*-value	<0.05	<0.05	
History pre (SD)	1.44 (0.50)	1.43 (0.54)	0.91
History post (SD)	1.44 (0.50)	1.43 (0.50)	0.91
*p*-value	<0.05	<0.005	
Time pre (SD)	1.86 (0.59)	1.83 (0.60)	0.86
Time post (SD)	1.16 (0.73)	1.13 (0.71)	0.85
*p*-value	<0.05	<0.05	
Failure pre (SD)	2.30 (0.46)	2.29 (0.46)	0.93
Failure post (SD)	0.41 (0.50)	0.43 (0.50)	0.89
*p*-value	<0.05	<0.05	
Assistance pre (SD)	1.08 (0.69)	1.18 (0.39)	0.42
Assistance post (SD)	0.63 (0.48)	0.56 (0.50)	0.54
*p*-value	<0.05	<0.05	
Pain pre (SD)	1.47 (0.99)	1.49 (0.98)	0.95
Pain post (SD)	0.50 (0.50)	2.00 (0.97)	<0.05
*p*-value	<0.05	<0.05	
Total score pre (SD)	13.9 (1.8)	13.72 (1.92)	0.87
Total score post (SD)	6.88 (1.63)	8.56 (1.76)	<0.05
*p*-value	<0.05	<0.05	

SD, standard deviation; pre, preoperatively; post, postoperatively.

## Data Availability

The data that support the findings of this study are restricted by privacy but might be available upon reasonable request.

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
