# Peer review of "Robot-Assisted Sacrocolpopexy versus Trans-Vaginal Multicompartment Prolapse Repair: Impact on Lower Bowel Tract Function"

_biomedicines, 2023, doi:10.3390/biomedicines11082105_

Round 1

Reviewer 1 Report

I am very much in support of your hypothesis that pelvic floor disorders should be diagnosed and treated with RSC. Unfortunately, your data and choice of operative approach cannot allow for any logical conclusions. The inclusion and exclusion criteria as well as the primary and secondary outcome measures should have been specified.   What score did the authors consider on the Wexner scale to state that the patients had defecation disorders? Evaluation of obstructed defecation is multimodal. Additional modalities to elicit the diagnosis of obstructed defecation include proctoscopy, colonic transit time studies, anorectal manometry, a rectal balloon expulsion test, defecography, electromyography, colonoscopy and ultrasound. The lack of power calculation is noted. Continuous variables should been checked for normality before deciding whether to use parametric or non-parametric tests for their description and comparison. Descriptive statistical tests should have been outlined. The range has no place in statistics. Avoid all conclusions  that are not supported by the data 

Moderate editing of English language

Author Response

Reviewer#1 

I am very much in support of your hypothesis that pelvic floor disorders should be diagnosed and treated with RSC. Unfortunately, your data and choice of operative approach cannot allow for any logical conclusions. 

The inclusion and exclusion criteria as well as the primary and secondary outcome measures should have been specified.

Answer: Thanks for this important recommendation, we went through the text file, and according to your suggestion, we completely rewrite the inclusion exclusion criteria as well as the outcome measures.      

What score did the authors consider on the Wexner scale to state that the patients had defecation disorders? Evaluation of obstructed defecation is multimodal. Additional modalities to elicit the diagnosis of obstructed defecation include proctoscopy, colonic transit time studies, anorectal manometry, a rectal balloon expulsion test, defecography, electromyography, colonoscopy, and ultrasound.

Answer: This is a very important information that was not clearly and adequately expressed in the previous main text. We did add, following your kind requests, all the modalities used to evaluate the ODS. First, we detailed the self-administered questionnaire used in the study (questionnaire Agachan-Wexner scoring system, reference no.10), then, ODS was investigated by a coloproctologist, who was able to identify the presence of rectocele and external/internal prolapse with a proctological examination. Furthermore, all patients underwent defecography, anoscopy, anorectal manometry and electromyography to assess defecatory dynamics and to evaluate rectal prolapse, and colonoscopy, to exclude cancer or polyps.

The lack of power calculation is noted. Continuous variables should be checked for normality before deciding whether to use parametric or non-parametric tests for their description and comparison. Descriptive statistical tests should have been outlined. The range has no place in statistics. Avoid all conclusions that are not supported by the data. 

Answer: Thank you for your careful observation. In relation to the samples analyzed, 2 groups each with n > 30 patients, and the reduced number of studies that correlate these two procedures, we opted for the use of parametric tests. However, non-parametric tests were also performed to confirm the results obtained although not reported in the study. 

Reviewer 2 Report

In my opinion, the analyzed topic is interesting enough to attract the readers’ attention. This study evaluated the effectiveness, safety, and possible changes in bowel symptoms after multi-compartment prolapse surgery comparing two different surgical approaches, transvaginal mesh surgery with levatorplasty and robot-assisted sacrocolpopexy.

I think that the abstract of this article is very clear and well structured.

Maybe, it could be useful the evaluation of a bigger study in order to compare better the surgical approaches. Also, possible urinary consequences should be analyzed. In particular, I suggest these articles to get deeper in the topic: PMID: 28786873 and PMID: 32744453. Because of these reasons, the article should be revised and completed. The tables and figures are very interesting. Considered all these points, I think it could be of interest for the readers and, in my opinion, it deserves the priority to be published after revisions.

a moderate review of english is necessary.

Author Response

Reviewer#2

In my opinion, the analyzed topic is interesting enough to attract the readers’ attention. This study evaluated the effectiveness, safety, and possible changes in bowel symptoms after multi-compartment prolapse surgery comparing two different surgical approaches, transvaginal mesh surgery with levatorplasty and  robot-assisted sacrocolpopexy.

I think that the abstract of this article is very clear and well structured.

Answer: Thanks for your comment we do appreciate your evaluation on our study.

Maybe, it could be useful the evaluation of a bigger study in order to compare better the surgical approaches. Also, possible urinary consequences should be analyzed. In particular, I suggest these articles to get deeper in the topic: PMID: 28786873 and PMID: 32744453. Because of these reasons, the article should be revised and completed. The tables and figures are very interesting. Considered all these points, I think it could be of interest for the readers and, in my opinion, it deserves the priority to be published after revisions.

Answer: Thanks for your precious suggestion, according to your comment we added to the discussion these two studies and we completed the section, adding these data of the two studies.

Round 2

Reviewer 1 Report

The authors changed the observation period of the study from 2021 to 2022. They do not report any data from the preoperative investigations carried out (anorectal manometry, defecography, etc.). I consider the review insufficient

minor revision

Author Response

The authors changed the observation period of the study from 2021 to 2022.

Answer: Sorry for this typing error, the correct year is 2021 as in the original draft.

They do not report any data from the preoperative investigations carried out (anorectal manometry, defecography, etc.). I consider the review insufficient

Answer: According to your comment we did add these missing data in table 1, reporting all the preoperative investigations performed. These missing data are referred to the anal sphincter manometry performed preoperatively in rest and squeezing conditions, reporting no significant differences between the two groups (p > 0.05). Regarding MRI defecography, as described in the methods section, all patients underwent MRI defecography to assess defecatory dynamics and to evaluate rectal prolapse, and colonoscopy, to exclude cancer or polyps. No statistically significant differences were observed at MRI between the two groups. All patients reported a multicompartmental prolapse with rectocele, excluding other anorectal radiological abnormailities.

Reviewer 2 Report

I read with great interest the Manuscript titled " Robot-assisted sacrocolpopexy versus trans- 2

vaginal multicompartment prolapse repair: im- 3 pact on lower bowel tract function.”, topic interesting enough to attract readers' attention. 

The quality of the manuscript has improved thanks to the changes made. The methodology is accurate, and results are supported by data analysis. The discussions are clear and comprehensive. 

In the "Authors' response" section, I read that the authors added two suggested studies (PMID: 28786873 and PMID: 3274445) to the discussion, and completed the section. 

But I recommend adding these two references (PMID: 28786873 and PMID: 32744453) also in the References section because do not appear.

Because of these reasons, the article should be revised and completed. Considering this point, I think it deserves priority to be published after minor revisions.

Author Response

I read with great interest the Manuscript titled " Robot-assisted sacrocolpopexy versus trans-vaginal multicompartment prolapse repair: impact on lower bowel tract function.”, topic interesting enough to attract readers' attention. 

The quality of the manuscript has improved thanks to the changes made. The methodology is accurate, and results are supported by data analysis. The discussions are clear and comprehensive. 

In the "Authors' response" section, I read that the authors added two suggested studies (PMID: 28786873 and PMID: 3274445) to the discussion, and completed the section. But I recommend adding these two references (PMID: 28786873 and PMID: 32744453) also in the References section because do not appear.

Because of these reasons, the article should be revised and completed. Considering this point, I think it deserves priority to be published after minor revisions.

Answer: thanks for all your considerations, unfortunately we did not add the 2 references in the previous text file, now we have completed the reference list revision and we added these two papers to the final references.

Round 3

Reviewer 1 Report

 I consider the review sufficient

Minor editing of english required